# Psychobiotic Properties of *Lactiplantibacillus plantarum* in Neurodegenerative Diseases

**DOI:** 10.3390/ijms25179489

**Published:** 2024-08-31

**Authors:** Mariagiovanna Di Chiano, Fabio Sallustio, Daniela Fiocco, Maria Teresa Rocchetti, Giuseppe Spano, Paola Pontrelli, Antonio Moschetta, Loreto Gesualdo, Raffaella Maria Gadaleta, Anna Gallone

**Affiliations:** 1Department of Translational Biomedicine and Neuroscience, University of Bari Aldo Moro, Piazza Giulio Cesare n. 11, 70124 Bari, Italy; mariagiovanna.dichiano@uniba.it (M.D.C.); anna.gallone@uniba.it (A.G.); 2Department of Precision and Regenerative Medicine and Ionian Area, University of Bari Aldo Moro, Piazza Giulio Cesare n. 11, 70124 Bari, Italy; fabio.sallustio@uniba.it (F.S.); loreto.gesualdo@uniba.it (L.G.); 3Department of Clinical and Experimental Medicine, University of Foggia, 71122 Foggia, Italy; daniela.fiocco@unifg.it (D.F.); mariateresa.rocchetti@unifg.it (M.T.R.); 4Department of Agriculture Food Natural Science Engineering (DAFNE), University of Foggia, 71122 Foggia, Italy; giuseppe.spano@unifg.it; 5Department of Interdisciplinary Medicine, University of Bari Aldo Moro, Piazza Giulio Cesare n. 11, 70124 Bari, Italy; antonio.moschetta@uniba.it (A.M.); raffaella.gadaleta@uniba.it (R.M.G.); 6National Institute for Biostructure and Biosystems (INBB), Viale delle Medaglie d’Oro n. 305, 00136 Roma, Italy

**Keywords:** microbiota, microbiome, neurological disorders, *Lactiplantibacillus plantarum*, gut–brain axis

## Abstract

Neurodegenerative disorders are the main cause of cognitive and physical disabilities, affect millions of people worldwide, and their incidence is on the rise. Emerging evidence pinpoints a disturbance of the communication of the gut–brain axis, and in particular to gut microbial dysbiosis, as one of the contributors to the pathogenesis of these diseases. In fact, dysbiosis has been associated with neuro-inflammatory processes, hyperactivation of the neuronal immune system, impaired cognitive functions, aging, depression, sleeping disorders, and anxiety. With the rapid advance in metagenomics, metabolomics, and big data analysis, together with a multidisciplinary approach, a new horizon has just emerged in the fields of translational neurodegenerative disease. In fact, recent studies focusing on taxonomic profiling and leaky gut in the pathogenesis of neurodegenerative disorders are not only shedding light on an overlooked field but are also creating opportunities for biomarker discovery and development of new therapeutic and adjuvant strategies to treat these disorders. *Lactiplantibacillus plantarum* (LBP) strains are emerging as promising psychobiotics for the treatment of these diseases. In fact, LBP strains are able to promote eubiosis, increase the enrichment of bacteria producing beneficial metabolites such as short-chain fatty acids, boost the production of neurotransmitters, and support the homeostasis of the gut–brain axis. In this review, we summarize the current knowledge on the role of the gut microbiota in the pathogenesis of neurodegenerative disorders with a particular focus on the benefits of LBP strains in Alzheimer’s disease, Parkinson’s disease, amyotrophic lateral sclerosis, autism, anxiety, and depression.

## 1. Introduction

Neurodegenerative diseases represent a serious global burden often associated with aging. In 2019, almost 350 million neurological disease cases, especially Alzheimer’s and Parkinson’s diseases (AD and PD, respectively), and over 10 million associated deaths were reported [1]. Aging is a physiological process; however, its effects can promote cognitive decline and oxidative stress. Classically, the relationship between neurology and microbiology has been ascribed to prion infections, hepatic encephalopathy, sepsis, and viral or bacterial infection of the central nervous system (CNS). The blood–brain barrier (BBB) regulates the selective entry of molecules into the brain [2], protecting the CNS from the entrance of harmful triggers. Strokes and neurodegenerative diseases have been associated with a breakdown of the BBB [3]. As a result, pathogens can cross the BBB and trigger infections that cause neuroinflammation, such as meningitis, encephalitis, and focal abscesses, causing debilitating effects and in some cases even death [3]. In the last two decades, research breakthroughs have shown that the microbiota resident in the gastrointestinal (GI) system has a major role in regulating the communication between the gut and the CNS. In fact, when perturbation of the intestinal homeostasis occurs, not only can the function of the GI tract be compromised but also the brain can be heavily affected.

GI physiological activities range from food ingestion to nutrient digestion and absorption to metabolic activities. Nutritional- and nervous-dependent regulatory mechanisms ensure maximum exposure of nutrients to the mucosa of the small intestine, responsible for nutrient absorption [4]. The intestinal mucosa represents the interface between the host and the external environment and is exposed to a wide range of antigens. Therefore, the gut also serves as a first line defense against potential harmful triggers and assumes vital roles in the formation and training of important innate and adaptive immune system components in the host. About 70% of all immune cells and antibodies pass through or mature in the Peyer’s patches of the gut. 

The gut microbiota (GM) consists of over 40 trillion microbial organisms, including bacteria, viruses, protozoa, and yeasts [5]. The bacterial domain is the most represented as it is the most abundant and diverse [6]. To date, nine different bacterial phyla have been identified. Among these, *Firmicutes* and *Bacteroidetes* are the most abundant, followed by *Actinobacteria* and *Proteobacteria* [7,8]. The GM is essential for our life and health, and its composition dynamically shifts over the human lifespan. It changes over time, from the moment a baby is born, through childhood and adulthood, and in elderly life [9]. During the first year of life, *Bifidobacteria* stimulate the activation of the metabolism of human milk oligosaccharides, bringing potentially beneficial effects, such as an increase in the production of type A immunoglobulins and the strengthening of the intestinal mucosa barrier [10]. Subsequently, through the initial introduction of food and during adulthood, the composition of the GM keeps shifting [11].

In physiological conditions, gut microbial composition is in the so-called eubiosis, with a preponderance of species that have beneficial potential. On the contrary, dysbiosis represents the disruption of this balance. GM shifts are influenced by nutritional choices, lifestyle, drug use, and exposure to different environmental factors [12]. The most classical causes of dysbiosis are antibiotic treatment or intestinal infections [13,14], primarily causing a GM compositional shift resulting in the loss of bacterial diversity with a decrease in beneficial species and the establishment of a niche more favorable to pathogens, which in turn disrupts the intestinal barrier integrity and primes inflammation [13]. Moreover, upon dysbiosis, immunomodulatory functions can be impaired [13]. Given the tight connection between the gut and the rest of the body, the consequences of dysbiosis can be observed not only in the intestine itself but also on distant organs. In the last two decades, the development of next-generation sequencing techniques, whole-genome shotgun sequencing, global metabolomics, and advanced computational strategies, along with humanized animal models and culture-based human organoid systems, has aided a first understanding of the GM and its functions not only in the gut but also in its interaction with other systems [15].

Gut microbes are made up of genes that code for thousands of microbial enzymes and produce a myriad of metabolites [16,17], which play fundamental roles ranging from the regulation of digestive and absorptive processes to energy harvesting [17], metabolism [17], and even in the activation of the immune and nervous systems [16,17]. In fact, microbial-derived metabolites, generated according to the substrate or nutrient they feed on, are able to reach different organs and modulate their function. One of the most important sets of microbial metabolites, produced upon fiber fermentation, are short-chain fatty acids (SCFAs), namely, butyrate, acetate, and propionate. Amongst other functions, butyrate represents a fundamental energy source for colonocytes [18]; acetate is involved in the modulation of body weight maintenance through different mechanisms affecting central appetite regulation, gut-satiety hormones, and improvements in lipid metabolism and energy expenditure [18]; and propionate is involved in lipid and glucose metabolism [19]. All of them have putative health effects that extend beyond the gut epithelium and are main players in the mutual communication of the gut–other organs axes, such as the gut–liver and the gut–brain axes. Moreover, bacteria can produce some vitamins, tryptophane, and polyphenols metabolites [20,21].

## 2. The Gut–Microbiota–Brain Axis in Health and Neurodegeneration

For all these reasons, the GM is at the crossroad of multiple interactions within the host, including the gut–brain axis, a dynamic system characterized by a bidirectional communication via microorganisms’ and their byproducts’ passage from the gut to the brain and the parasympathetic nervous system innervating the bowel [22,23], linking the gut with the emotional and cognitive centers of the brain. The BBB is a fundamental part of this communication network. Impairment of the BBB physiology can increase the susceptibility to neurodegenerative disorders such as AD, PD, multiple sclerosis (MS), and cerebrovascular disease [24]. Despite being considered impermeable, the BBB permits the passage of neurotransmitters, immune-competent cells, and certain bacterial metabolites. In the brain, GM metabolites are involved in the activation of neuroprotective systems and contribute to priming the production of serotonin, dopamine, antioxidant enzymes, and regulatory proteins of cellular calcium homeostasis [25].

Recent studies have identified the so-called “neuro-metabolites”, molecules secreted directly by the GM or secretory intestinal epithelial cells stimulated by the microbiota, exerting direct actions targeting the central nervous system (CNS). These molecules, including neurotransmitters, directly or indirectly influence the CNS, triggering the activation of numerous signalling pathways [22,23]. *Bifidobacteria*, *Lactobacillus*, *Streptococcus*, and *Enterococcus* spp. produce serotonin [26]. In fact, up to 95% of serotonin is produced by the gut-microbial-dependent metabolization of tryptophan [27]. Serotonin helps coordinate brain functions and affects heart function, bowel motility, ejaculatory latency, bladder control, and platelet aggregation. *Lactobacillus* and *Bifidobacteria* are able to secrete gamma amino-butyric acid (GABA), which displays an inhibitory effect of a nerve impulse at the postsynaptic level on mammals’ CNS [28,29,30,31]. *Bifidobacterium dentium*, *Bifidobacterium longum subsp. Infantis* [30,32], and *Bifidobacterium adolescents* [33] have been shown to produce GABA in vivo and have recently been designated with the name of “psychobiotics”, i.e., capable of influencing neurological activities such as sleep, appetite, mood, and cognition by modulating neuronal signals [29]. Moreover, it has been observed that *Lactiplantibacillus plantarum* (LBP) PS128 improves atypical behaviors and determines the regulation of both the dopaminergic and serotonergic signalling pathways in the brain of mice. In particular, studies on a GF mouse model [34] and early-life-stressed and naïve adult mice [35] have shown that the administration of LBP PS128 would induce emotional changes and consequently behavioral changes associated with an increase in the levels of monoamine neurotransmitters. 

In the CNS, SCFAs not only contribute to maintaining the integrity of the BBB but also collectively influence behavior, memory, synaptic plasticity, learning, and other neurological function. Germ-free (GF) mice display, in fact, increased gut and BBB permeability, and it has been shown that supplementation with the SCFAs-producing *Clostridium* tyrobutyricum restores both gut and brain barrier homeostasis [36]. SCFAs also influence the production of glutamate, glutamine, GABA, and neurotrophic factors. Propionate and butyrate modulate the expression of serotonin- and catecholamine-synthesizing enzymes and regulate intracellular potassium levels (reviewed in [37]). In conditions of dysbiosis, some pathogens could produce metabolites, such as D-lactic acid, ammonia, and indoxyl sulphate, that can exert neurotoxic effects, impairing cognitive functions and functional response of the adaptive immune system in the brain [38,39,40,41,42]. Furthermore, the current knowledge of the intestinal microbiota in the context of depression has led to highlight the importance of SCFAs in counteracting this pathology. In particular, some studies on mouse models with depression have shown that butyrate has antidepressant properties counteracting the behavioral alterations associated with cognitive and social disorders [43,44,45].

The GM also closely interacts with other important components involved in neuroprotection, and, in particular, homeostasis of glia cells has been shown to be influenced by the GM [46]. Glia cells are non-neuronal cells located in both the central and the peripheral nervous system. They have several functions, ranging from maintenance of the nervous system homeostasis to myelin formation, oxygen and nutrient supply to neurons, and pathogen disruption. Enteric glial cells are present in the intestinal mucosa directly beneath enterocytes and are responsible for neuroprotection, maintenance of the intestinal barrier, and modulation of the immune response [47]. It has been observed that the GM is involved in the postnatal developmental migration process of glial cells [46] and controls maturation and functions of microglia in the nervous system [46,48]. Homeostasis of glial cells has been shown to be influenced by the GM [46], and, in particular, the glia would not be able to respond to immunostimulant agents such as lipopolysaccharides (LPS) or viruses in the absence of microbiota [49].

So far, the evidence discussed strongly indicates that a molecular miscommunication in the gut–brain axis contributes to neuropsychiatric and neurodegenerative diseases (Figure 1). Intestinal inflammation impairs the intestinal barrier integrity, thereby increasing mucosal permeability. This leads to leakage of pathogens and their metabolites into systemic circulation, which can elicit immune signalling and inflammation. Emerging evidence points to gut dysbiosis as a contributor to the onset and development of neurodegenerative diseases, such as AD, PD, amyotrophic lateral sclerosis (ALS), and multiple sclerosis (MS) [50,51] (Figure 2). For instance, major changes in GM composition, together with quickened aging and neurodegenerative hallmarks such as tau phosphorylation, β-amyloid formation, and neuroinflammation, have been observed in patients affected by AD [52,53]. Furthermore, dysbiosis could aggravate neurodegeneration through harmful triggers, such as altered plasma and colonic metabolic profile of cerebral neuropeptide Y, thereby contributing to brain aging, depression, sleeping disorders, and anxiety [54,55,56]. Last but not least, an aging brain can generally be characterized by a compromised DNA repair system, mitochondrial dysfunctions, inflammation, oxidative damage, autophagy, dysregulated neuronal calcium homeostasis, and an abnormal neuronal network, all together escalating the susceptibility to AD, PD, and stroke [57].

Conversely, the promotion of a healthy GM may minimize the risk of neurodegenerative disease [58,59,60]. Emerging evidence shows that the administration of adjuvant probiotics, live bacteria that in adequate quantities are able to confer a health benefit to the host organism [61], represent a common promising practice able to restore eubiosis in several conditions, including aging [62,63,64,65]. In particular, the most recent data indicate that the probiotic bacterium LBP is not only beneficial in chronic inflammatory diseases of the gut, cancer, infections, and pregnancy [66,67,68] but also in the gut–brain axis, preserving the BBB integrity and counteracting neurodegeneration in both rodent models and humans [69,70,71]. Several (mostly) preclinical studies have also shown that different LBP strains are also able to counteract neuronal oxidative stress and cognitive decline in aging and AD [72,73,74,75,76,77,78,79,80,81,82,83].

## 3. Lactic Acid Bacteria (LAB), *Lactiplantibacillus plantarum*, and Neurodegenerative Diseases

Lactic acid bacteria (LAB) are a heterogeneous vast group of Gram-positive bacteria. Thanks to their metabolic and safety features, since ancient times, several LAB have been used to ferment food, contributing to its preservation and sensory and nutritional quality; in the last decades, various species have been drawing scientific interest for their probiotic properties [84]. LAB display a plethora of putative health benefits ranging from enhancement of lactose digestion and control of intestinal infections to immune system regulation and preservation of the gut. Lactic acid is their main catabolic product, resulting from carbohydrate fermentation [85]. Within the LAB group, the *Lactobacillus* genus is the most represented yet heterogeneous genera, consisting of over one hundred identified species with substantial differences in their genotypic, phenotypic, and physiological characteristics [85]. *Lactobacilli* strains are widespread in nature and have many health-promoting activities and a long, safe history of being consumed by humans [86]. One of the most important activities of commensal bacteria is the promotion of hosts’ health, achieved by modulating the mucosal immune system. LAB-dependent immunomodulatory activities are not only essential for the activation of tolerogenic mechanisms to foreign harmless antigens [87] but also crucial for the maintenance of intestinal homeostasis [88]. *Lactobacilli* have also been shown to have antioxidant properties, possibly ascribable to their capacity of producing antioxidant metabolites and enzymes to scavenge ROS, upregulating hosts’ antioxidant enzymes activities while inhibiting enzymes ROS production and regulating antioxidant-related signalling pathways of the host and the host’s GM [89,90,91].

The dietary intake of probiotic supplements and probiotic-containing products, such as yogurts and fermented food, is emerging as an effective strategy to beneficially manipulate the GM composition, resume eubiosis, and increase diversity. *Lactobacilli*, often in combination with *Bifidobacteria*, have been shown to promote immunomodulatory functions, produce microbial byproducts such as short-chain fatty acids (SCFAs) and colonic immunoglobulin A (IgA), and regulate GI functions. Among the *Lactobacillus* genus, *Lactiplantibacillus plantarum* (LBP), a heterofermentative, nonmotile, non-spore-forming bacterium, is one of the most promising species displaying beneficial effects on health [92]. LBP is very widespread in the environment, as it colonizes soil, vegetable-related, and food-related niches, including fermented food for human consumption. Moreover, LBP is a natural inhabitant of human mucosae, including those in the mouth and vaginal and intestinal tracts [93,94]. LBP strains with probiotic claims are currently commercialized in the form of dietary supplements and diverse probiotic formulations [71,95].

Harnessing the gut microbiota involves several strategies encompassing the use of probiotics, prebiotics, and symbiotics; antibiotics; and fecal microbiota transplantation (FMT). A substantial body of evidence confirmed the successful use of diverse human probiotic LBP strains as a dietary intervention to prevent and/or ameliorate some pathological conditions, such as cardiovascular diseases [96], GI infections [97,98], gynecological diseases [99], irritable bowel syndrome and inflammatory bowel disease [100,101], colorectal cancer [102], hypercholesterolemia and obesity [103,104], and diabetes [105]. For these reasons, LBP strains are being used in clinical studies in both diseased and healthy subjects. For instance, in a randomized, placebo-controlled, double-blinded crossover trial, 22 healthy subjects underwent four-week treatment periods with either a mixture containing LBP R1012, *Bifidobacterium longum* R0175, and *Lactobacillus helveticus* R0052 or a placebo, separated by a four-week washout period, and probiotic supplementation intervention evoked distinct changes in brain morphology and resting state brain function, alongside improvements of psycho(bio)logical markers of the gut–brain axis [106]. In this context, LBP supplementation is emerging as an adjuvant strategy for the clinical management of patients with cognitive impairments [107] and neurodegenerative diseases (Table 1).

LBP psychobiotic activities are thought to go through the activation of various mechanisms, such as GM reshape, activation of the Nuclear factor-E2-related factor 2 (Nrf2), decreased production or enhanced scavenging of ROS, and production of anti-inflammatory cytokines [130]. The beneficial reshape of the GM aids intestinal motility and eases the production of mucus, SCFAs, and neurotransmitters such as GABA, which plays a major role in reducing anxiety and pain in the nervous system, and serotonin [113]. Moreover, via the inhibition of metabolites such as kynurenine, the GM is able to promote anti-inflammatory mechanisms and antioxidant machinery and boost neurogenesis [107,131,132,133]. Amongst all their physiological functions, SCFAs are also able to activate Nrf2 [134], which in turn activates DNA repair mechanisms and regulates a plethora of antioxidant genes in the striatum cells, brain, and colon [109,125]. Furthermore, LBP has been shown to inhibit hyperactivation of the microglia, which are considered key immune cells of the CNS [135]. When the physiology of the microglia is altered, a hyperactivation of the neural immune response and inflammation occur, leading to neurodegeneration [135]; in this respect, LBP is able to reduce proinflammatory cytokine production while promoting anti-inflammatory mechanisms in the microglia [135].

In the following sections, we review the evidence for the role of the GM and discuss the potential psychobiotic role of LBP supplementation in neurodegenerative disorders.

## 4. Alzheimer’s Disease, Cognitive Impairment, and Aging

AD or cognitive impairment is a typical pathology of elderlies affecting the CNS and characterized by a progressive cognitive decline. It has been estimated that the current number of people affected by Alzheimer’s dementia will more than double by 2060, and AD will represent the sixth-leading cause of death in the United States [136,137]. One of the cornerstones of aging is the loss of GM diversity of important taxa, such as *Bacteroides*, *Prevotella*, and *Lactobacilli*, with a concomitant increased abundance of *Ruminococcus* and *Enterobacteriaceae* [136]. A main feature of AD is an excessive deposition of amyloid-β (Aβ) and hyperphosphorylated tau [138], fundamental structural proteins of extracellular senile plaques and intracellular neurofibrillary tangles, respectively. The relation between amyloid accumulation and neuroinflammation leading to loss of synapses and cognitive decline are still under debate. However, emerging observations point to the involvement of a dysbiotic GM as a pathogenetic contributor in AD. Bacterial endotoxins have been previously shown to be involved in amyloidosis and associated neuroinflammation in AD [139,140] and found as a component of plaques [139,141]. A study in over 200 AD patients demonstrated a fecal increased abundance of *Escherichia*/*Shigella*, i.e., taxa with proinflammatory abilities, and a reduced abundance of the beneficial *Eubacterium rectale* compared to control subjects [142]. In these patients, this peculiar GM signature was significantly associated with blood inflammatory biomarkers, such as interleukin-6 (IL-6), C-X-C motif chemokine ligand 2 (CXCL2), and the nucleotide-binding oligomerization domain (NOD) receptor and leucine-rich repeat and pyrin domain containing-3 (NLRP-3) [142]. Moreover, fecal fungal dysbiosis was observed in a Chinese cohort of AD patients, with an enrichment of *Candida tropicalis* and *Schizophyllum commune*, whose presence was also positively associated with IP10 and TNFα, and a decreased abundance of *Rhodotorula mucilaginosa*, negatively associated with TNFα [143]. 

Impaired levels of GM metabolic products, especially SCFAs, was also suggested as a contributing factor of AD pathogenesis. A lower GM diversity, paired with reduced circulating SCFAs levels, were found in an AD mouse model [144]. Metabolic prediction highlighted alterations in more than 30 metabolic pathways potentially associated with amyloid accumulation and intestinal morphological abnormalities in these mice [144]. Furthermore, it was observed that SCFAs are able to interfere with the formation of protein-protein bonds and between amyloid-beta (Aβ) peptides, thus blocking the production of neurotoxic oligomers responsible for synaptic dysfunction and cognitive disorders associated with AD [106,145]. The role of LBP supplementation in AD has the potential to beneficially influence the GM composition, circulating SCFAs levels, and inflammation, ultimately improving cognitive functioning in AD animal models [146,147]. Moreover, in an AD-induced rat model, the LBP MTCC1325 strain reverted all the constituents of ATPase enzymes, which are involved in neuronal energy metabolism and known to be involved in AD progression when their levels are reduced, and delayed neurodegeneration [122]. Another strain supplementation in AD mice, namely, LBP PS128, was suggested to improve motor function and to oppose cognitive decline, depression and anxiety behavioral features, Aβ deposition, fecal levels of the SCFA propionate, and other neurodegeneration markers [108].

Aging-associated neurodegeneration is known to also affect the left brain hemisphere, and areas with asymmetric gray matter decline were proposed to be associated with neurodegeneration [148]. Results of a recent clinical study analyzing the probiotic effect on this condition have shown that *Lactobacillus*-based probiotic supplements are able to decrease depressive symptoms and increase gray matter volume [149].

## 5. Parkinson’s Disease (PD)

PD is a neurodegenerative disease characterized by several motor and nonmotor symptoms accumulating over time [150] due to dopaminergic neuron loss in the substantia nigra, striatal dopamine deficiency, and accumulation of misfolded α-synuclein. Nowadays, PD has become one of the main causes of disability worldwide, which causes a significant burden on individuals and on the health care system [150,151]. PD mostly occurs in elderlies; however, it can also affect younger adults of less than 50 years of age. According to the latest data from the Parkinson’s Foundation, over 10 million people suffer from PD worldwide. Between 1994 and 2019, there was a significant increase in global mortality rates in both men and women, which rose from 1.76 in 1994 to 5.67 in 2029 (per 100,000 cases) [152]. Anyhow, the incidence of PD can differ in patients according to genetic and exposome factors. Higher mortality was observed in men than in women and in older than in younger people [152]. Frequent symptoms observed in PD patients are slowing of movements, tremor, lack of balance, rigidity, tremors, painful muscle contractions, and difficulty speaking. Also, a plethora of nonmotor conditions such as cognitive impairment, depression, autonomic dysfunction, and hyposmia have also been described. 

This pathological condition is not curable, but there are several treatment options that aim to slow down disease progression [153] and improve patients wellbeing, such as deep brain stimulation and dopamine substitution [154,155]. The gut microbiota–brain axis is a topic of active discussion in the field of neurodegenerative disease, including PD. Gut microbiota sequencing data have shown an enrichment of *Enterobacteriaceae* in the stools of patients with PD, directly correlated to the severity of postural instability [156]. Moreover, some bacterial species such as *Escherichia coli*, *Klebsiella*, *Salmonella*, *Shigella*, and *Yersinia pestis* produce proinflammatory LPS, positively associated with motor severity in PD subjects [156,157]. Dysbiosis, reduced abundance of butyrate-producers, and consequently lower SCFAs levels have been observed in PD patients, paired with increased levels of endotoxin and neurotoxin, both potentially linked to PD development. Lower SCFAs levels in PD patients have been significantly associated with poor cognition and low body mass index (BMI), and in particular, lower butyrate levels are directly correlated to postural instability–gait disorder scores [158]. SCFAs are also involved in GI motility and physiologically regulate the enteric nervous system; therefore, PD patients may present constipation [159]. Furthermore, an inverse correlation was observed between fecal SCFAs levels and several PD-related clinical variables such as the Non-Motor Symptoms Scale score, the Rome III constipation/defecation subscore, stool consistency associated with constipation on the Victoria Bowel Performance Scale, and the Geriatric Depression Scale-15 [160]. Supplementation of SCFAs in PD mice models, particularly butyrate, has been shown to improve motor functions and induces an increase in dopamine levels, suggesting that SCFAs may be a beneficial adjuvant treatment in the clinical management of PD [161,162]. LBP PS128 and CRL2130 probiotic supplementation has shown promising psychobiotic potential, especially in decreasing microglial activation, inflammation, neurotransmission, and neuronal death [60,109,163,164]. Clinical evidence also indicates that adjuvant LBP PS128 administration, in combination with antiparkinsonian medication, could improve the quality of life in PD patients [109].

## 6. Multiple Sclerosis (MS)

MS is a relapsing–remitting neuro-inflammatory disease caused by genetic and environmental factors, including intestinal dysbiosis. The disease commonly manifests by multiple demyelinating lesions in the white and gray matter of the brain and spinal cord, likely triggered by lymphocyte infiltration and antibody deposition promoting several neurological symptoms. Alterations of the GM, BBB, and T-cell-mediated autoimmunity have been shown to be important contributing factors in MS development [165]. To date, several studies have shown the role of the GM in MS in both human and rodent models. In particular, a reduced abundance of *Clostridium* clusters IV and XIVa, including *Faecalibacteriumprausnitzii*, *Eubacterium rectale*, and butyrate-producing bacteria, have been reported in MS patients [166], together with a reduction in other species, such as *Butyricimonas*, mainly belonging to *Bacteroides*, *Prevotella*, *Firmicutes*, and *Sutterella* [167,168]. Also, extensive studies in both animal models and humans strongly indicated the protective or pathogenic roles of GM in CNS autoimmunity [169,170,171,172,173,174], and that the GM and its metabolites affect the immune response and CNS resident cells including oligodendrocytes, astrocytes, and microglia [165,175]. From a therapeutic standpoint, it is known that the oral administration of SCFAs, particularly butyrate, in MS mouse models suppresses demyelination, inducing an improvement in oligodendrocyte remyelination [176,177].

As variation in the gut microbiome composition has been observed in MS patients, diet modulation or probiotic administration have both been trialed in MS patients. In a pilot study, Saresella et al. administered a high-vegetable/low-protein diet to a small cohort of MS patients, achieving clinical benefit [178], and observed an enriched relative abundance of a butyrate-producing bacterium belonging to *Firmicutes*. In another clinical trial, administration to MS patients of a probiotic mixture containing *Lactobacillus*, *Bifidobacterium*, and *Streptococcus* for two months restored *Lactobacillus* levels and reduced the immune response compared to the control group [110]. In a preclinical study, on a murine model of primary progressive MS, Vivomixx administration enriched the relative abundance of *Bacteroidetes*, *Tenericutes*, *Actinobacteria*, and *Saccharibacteria*; this was accompanied by a clear improvement of motor disability, decreased leukocyte infiltration, proinflammatory cytokine levels, microgliosis and astrogliosis, and increased plasma levels of butyrate and acetate [179]. In this context, LBP-based probiotic supplements have been shown to induce beneficial effects in MS when associated with other strategies. For instance, LBP probiotics combined with *Bifidobacterium animalis* were shown to reduce mononuclear infiltration in the CNS and improve MS-associated comorbidity as autoimmune encephalomyelitis in an MS model [123]. Furthermore, aerobic exercise paired with LBP oral gavage improved demyelination in the cuprizone-induced model of MS [127].

## 7. Amyotrophic Lateral Sclerosis (ALS)

ALS is characterized by progressive neurodegeneration, which causes a loss of motor neurons in the brain, brainstem, and spinal cord, which leads to a loss of voluntary skeletal muscle. Despite the very high clinical burden, there are no definite data about the global epidemiology of ALS; studies have been conducted on small datasets and are not multicentric, and therefore, this limits the general validity of the available information [180]. Recent data indicate that the prevalence of ALS in the US is estimated to be between 3.84 and 5.56 per 100,000 people, and the incidence is about 1.5 per 100,000 person-years [181]. Among the known symptoms of ALS, such as muscle weakness, muscle stiffness, and muscle spasms, there are also gastrointestinal symptoms, such as constipation, abdominal pain, a feeling of fullness, nausea, and difficult bowel movements, that seem to precede neurological ones [182]. It has been previously demonstrated that the SOD1^G93A^ ALS mouse model recapitulating the neuronal and muscle impairment of human ALS present with gut dysbiosis and increased intestinal permeability [183]. Also, feeding these mice with butyrate improves the intestinal barrier integrity and GM homeostasis, prolonging lifespan [184]. In line with this, in a small human study, Rowin et al. demonstrated that five patients with ALS and motor neuron disorder presented with an altered GM characterized by a lower diversity, low *Firmicutes*/*Bacteroidetes* ratio, and signs of GI disorders, such as gastroesophageal reflux, chronic constipation, bloating, intermittent diarrhea, and abdominal pain, when compared to control subjects [185]. In another study, Mazzini et al. studied the main human gut microbial groups and the overall microbial diversity in 50 ALS patients and 50 control subjects and also examined the effect of a mixture of *Lactibacillus* strains, including LBP. Preliminary results from this study indicate a difference in the GM composition in ALS patients, characterized by a higher abundance of *E. coli* and *Enterobacteria* and a low abundance of total yeast [116]. Furthermore, ALS subjects display a higher LPS level in plasma correlating to monocyte/macrophage activation, with higher LPS levels in patients with more severe disease [186]. Results of probiotic therapy are still in progress. 

## 8. Autism Spectrum Disorder (ASD)

Autism spectrum disorder (ASD) is a multifactorial neurodevelopmental disorder, influenced by genetic and environmental factors, characterized by deficits in social communication and the presence of restricted interests and repetitive behaviors, and estimated to have an international prevalence of 0.76%; however, this only accounts for approximately 16% of the global child population [187].

Treatments are multiple and not uniform, and further research is needed to understand behavioral and therapeutic response in ASD [188,189]. Although the topic has recently exploded and there is not enough knowledge or a universal consensus yet, dysbiosis has been directly or indirectly associated with ASD. In particular, a reduced [190] or excessive [191,192] abundance of lactobacilli has been reported in several studies and despite appearing contradictory [193], it suggests a clinical association between lactobacilli disbalance and the pathogenesis of ASD. The use of probiotics in the form of supplements was also described as an adjuvant therapeutic procedure to improve ASD-related symptoms and dysbiosis in affected children with an age ranging between 5 and 9 years old treated for 3 months with a multistrain probiotic containing *Lactobacillus acidophilus*, *Lactobacillus rhamnosus*, and *Bifidobacterium longum* [194]. Such improvements were also observed in a child who took multistrain supplements for 4 weeks [195]. From a neurophysiological point of view, ASD is characterized by an alteration of the GABAergic system, which is the main inhibitory neurotransmitter of the central nervous system. The atypical alternation between excitation and inhibition at the brain level affects the reactive behavior of ASD patients [196]. One study in particular highlighted that the integration of probiotics in ASD children promotes an improvement in brain functions, attenuating the imbalance of excitation/inhibition [196]. In autistic rats, the integration of probiotics, and in particular *B. longum*, CCFM1077, showed an improvement in brain activity and in the functionality of microglial cells present at the level of the cerebellum [197]. Data generated in murine models have shown that ASD symptoms could improve thanks to LBP supplementation by maintaining intestinal fitness and homeostasis [112,198]. Therefore, it seems that GM modulation could be helpful in some patients [199], as also shown in a randomized, double-blind, placebo-controlled trial in younger children within the spectrum [117]. Also, data generated in a study conducted in 35 subjects with ASD, which aimed to explore autonomic responses to daily LBP probiotic supplementation in combination with intranasal oxytocin, indicated that autonomic function indices improved (Table 2) [200]. 

## 9. Anxiety and Depression

As in the neurodegenerative disorders discussed so far, the GM could also affect behavior-related and psychiatric conditions [201]. As already mentioned, LBP is one of the most promising probiotic bacteria associated with CNS functions. It has been estimated that 25% of human drugs could negatively influence the GM, which could potentially even aggravate psychiatric disorders, and a large proportion of these medications are antipsychotics [202,203]. Probiotic supplementation may be a safe and effective therapeutic strategy in psychiatric disorders including anxiety and depression [204], and the clinical use of probiotics as a therapeutic strategy to alleviate depressive and anxiety-related symptoms has been considered effective not only in patients with active psychiatric disorders but also in healthy subjects subjected to very stressful daily life events [205]. Probiotic supplementation has been shown to increase the abundance of LBP involved in modulating psychiatric disorders and stress-induced behaviors [125,149,206]. Moreover, it has been shown in humans and mice that LBP strain PS128 beneficially modulates neurochemicals and stress-related symptoms, anxiety, depression-like behaviors, and major depression disorder [35,113]. Also, a very recent study in BALB/c mice indicated that LBP APsulloc 331,261 downregulated hippocampal inflammation, induced a microbial shift towards communities producing the SCFAs acetate and propionate, and attenuated depressive-like behavior [118]. Authors concluded that this modulation could stimulate an increase in colonic serotonin synthesis and neurogenesis in the brain [118]. In a double-blind, placebo-controlled trial, LBP 299v supplementation, in combination with selective serotonin reuptake inhibitors, in patients with a major depressive disorder improved cognitive performance and decrease kynurenine concentration (Table 2) [107].

## 10. Conclusions, Limitations, and Future Perspective

In recent years, much evidence has emerged from human and rodent models pointing to the role of the GM in the physiology of the CNS. Dysbiosis, in fact, has been strongly suggested to play a crucial part in the onset and development of neurodegeneration. In this review, we discussed the role of the GM in the health and diseases of the CNS, and in particular the psychobiotic potential of LBP in neurodegenerative disorders. Thanks to its ability to beneficially modulate GM composition by promoting the enrichment of SCFAs-producers and its capacity to boost the production of neurotransmitters, LBP supports the homeostasis of the gut–brain axis and regulates anti-inflammatory and antioxidant pathways in the CNS. All the emerging data are also laying the foundation for the design of novel therapeutic or adjuvant strategies for the clinical management of patients affected by neurodegenerative diseases. However, it is worth pointing out that we are far to make definitive conclusions, and caution should be taken in prescribing probiotic therapy in an indiscriminate way, as some groups have reported negative effects spanning from immunoreactivity to sepsis and even antibiotic resistance [207]. This should be considered especially in vulnerable subgroups, such as elderlies and immunocompromised and/or critically ill patients [208]. Moreover, we should be careful in translating data generated in animal models to humans, and even though data generated by RCTs are emerging, there are important pitfalls to take into account, such as length of treatment, strain of probiotics used, and dosage; moreover, patients’ recruitment in different RCTs, even within the same neurodegenerative disorder, are not optimal yet. In addition, to have more reliable data on intestinal colonization and benefits in patients’ physiology regarding the modulation of their GM, the formulation of probiotics administered to patients should be standardized to rely on the bioavailability of each strain or multistrains. Also, the definition of “healthy microbiome” is still one of the most enigmatic issues; more data on the role of diet-dependent GM changes in health and disease are needed; the interaction between drugs and the GM needs further studies; and there is no one-size-fit-all solution to target neurodegenerative disorders, despite the promising results. The diet and lifestyle of patients involved in these types of studies should always be considered, as they may influence intestinal colonization. For example, in AD, probiotics do not seems to establish colonization in a stable gut milieu; therefore, it becomes almost impossible to understand which particular bacteria in a multistrain formulation can be the most impactful [209]; moreover, in some AD patients, serotonin syndrome in subjects can occur [210]. Therefore, clinical trials design must be implemented, extending the number of recruited patients and foreseeing longer follow-up; furthermore, more endpoints should be included such as inflammatory biomarkers and specific cognitive assessment related to each neurodegenerative disease. Anyhow, the integration of multiomic data and an approach based on system biology will definitely move the field forwards, and more advanced data interpretation will allow a deeper comprehension of the conundrum of the gut microbiota–brain axis.

## Figures and Tables

**Figure 1 ijms-25-09489-f001:**
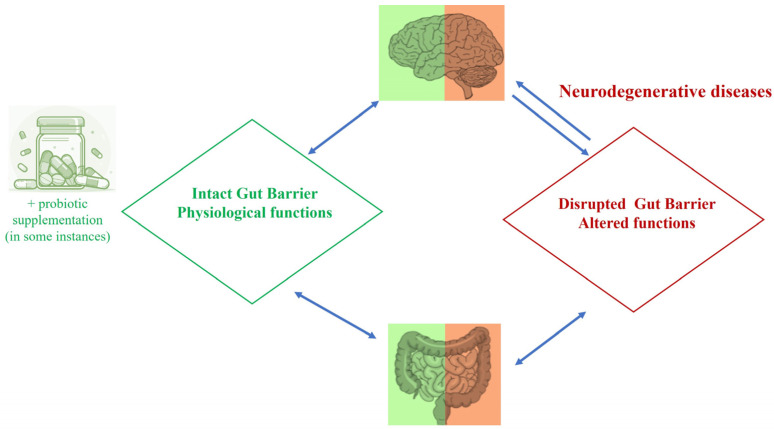
Caption. Microbiota-gut-brain-axis. Both eubiosis and an intact gut barrier promote a physiological communication within the gut-brain axis. On the contrary, a dysbiosic GM is associated with a disrupted intestinal barrier and triggers the release of inflammatory mediators that, one reaching the brain, cause neuronal changes leading to the pathogenesis of neurodegenerative disorders.

**Figure 2 ijms-25-09489-f002:**
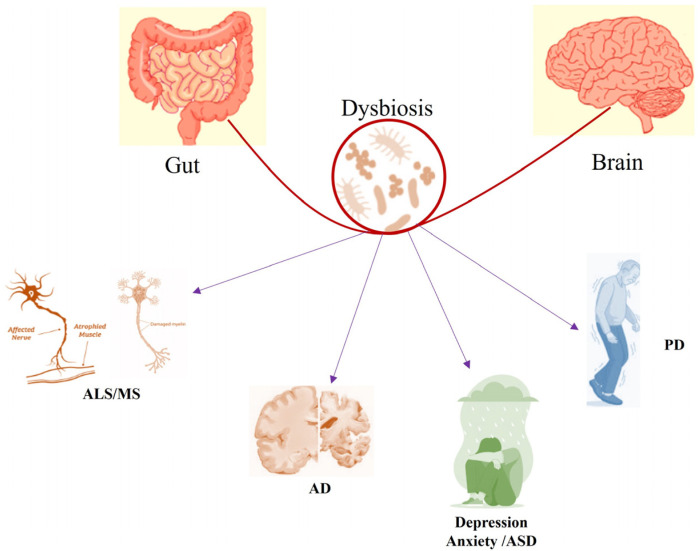
Caption. Dysbiosis affecting the gut-brain interaction. A Dysbalance of species abundance in the GM influences the gut-brain axis and plays a crucialrole in the pathogenesis of neurodegenerative disorders. Abbreviation: ALS amyotrophic lateral sclerosis, MS multiple sclerosis, AD Alzheimer disease, ASD autism spectrum disorder, PD Parkinson disease.

**Table 1 ijms-25-09489-t001:** LAB studied to attenuate neurodegenerative diseases. Among LAB, different species have been associated with neurodegenerative diseases. *Lactiplantibacillus plantarum* is one of the most studied in every neurodegenerative disease.

Alzheimer’s Disease	Parkinson’s Disease	Multiple Sclerosis	Amyotrophic Lateral Sclerosis	Autism Spectrum Disorder	Anxiety and Depression
*Lactiplantibacillus plantarum* R1012 [108]	*Lactiplantibacillus plantarum* PS128 [109]	*Lactiplantibacillus plantarum* DSM 24730 [110]	*Lactiplantibacillus acidophilus* [111]	*Lactiplantibacillus plantarum* ST-III [112]	*Lactiplantibacillus plantarum strain* PS128 [113]
*Bifidobacterium longum* NK46 [114]	*Lactiplantibacillus plantarum* DP189 [115]	*Bifidobacterium longum* DSM 2436 [110]	*Lactobacillus fermentum* [116]	*Lactiplantibacillus plantarum* PS128 [117]	*Lactiplantibacillus plantarum* APsulloc 331261 [118]
*Lactobacillus helveticus* IDCC3801 [119]	*Lactiplantibacillus plantarum* NCIMB 30173 [120]	*Streptococcus thermophilus* DSM 2431 [110]	*Lactobacillus delbrueckii* [116]	*Lactiplantibacillus plantarum* WCFS1 [121]	*Lactiplantibacillus plantarum* 299v [107]
*Lactiplantibacillus plantarum* MTCC1325- [122]	*Lactobacillus rhamnosus* NCIMB 30174 [111]	*Bifidobacterium* *Animalis* PTCC 1631 [123]	*Lactobacillus salivarius* [116]	*Bifidobacterium lactis* Probio-M8 [124]	*Lactiplantibacillus plantarum* JYLP-326 [125]
*Lactiplantibacillus plantarum* DSM 32244 [126]	*Lactobacillus acidophilus* NCIMB 30175 [120]	*Lactiplantibacillus plantarum* 1058 [127]	*Bifidobacterium* [116]	*Lactobacillus rhamnosus* HN001 [128]	*Lactiplantibacillus plantarum* DR7 [129]

**Table 2 ijms-25-09489-t002:** LBP studied in clinical trials suggested to attenuate neurodegenerative diseases. Among LBP, different strains have been associated with neurodegenerative diseases.

*Lactiplantibacillus Strain*	*Neurodegenerative Disease*	*Protocol Summary*
*Lactiplantibacillus plantarum* PS128	Parkinson Disease	n participants = 25 Duration of treatment = 12 weeks
*Lactiplantibacillus plantarum 299v* [107]	Anxiety and Depression	n participants = 40 Duration of treatment = 8 weeks
*Lactiplantibacillus acidophilus*	Amyotrophic Lateral Sclerosis	n participants = 42 Duration of treatment = 12 weeks
*Lactiplantibacillus plantarum* DSM 24730	Multiple Sclerosis	n participants = 9 Duration of treatment = 2 months
*Lactiplantibacillus plantarum* NCIMB 30173	Parkinson Disease	n participants = 3 Duration of treatment = 48 h
*Lactiplantibacillus plantarum* MTCC1325	Alzheimer’s Disease	n participants = 3 Duration of treatment = 48 h
*Lactiplantibacillus plantarum* WCFS1 [200]	Autism Spectrum Disorder	n participants = 35

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
