# Peer review of "Psychobiotic Properties of Lactiplantibacillus plantarum in Neurodegenerative Diseases"

_ijms, 2024, doi:10.3390/ijms25179489_

Round 1

Reviewer 1 Report

Comments and Suggestions for Authors

The review addresses a topic of great current interest: the role of GM in the pathogenesis of neurodegenerative disorders. The review is well structured and written. It is exhaustive despite having to analyze many topics citing the various mechanisms underlying the development of neurodegeneration. The citations are sufficient and relevant.

 I would suggest, as changes to the text, an expansion of paragraphs 8 and 9 (ASD and Anxiety and Depression) which are less in-depth than the others. Furthermore, to facilitate reading, some figures could be added that emphasize the most important information both in the initial part relating to the Gut-GM-Brain Axis and in the second part of the review relating to the various neurodegenerative diseases.

Author Response

Comments 1: 

The review addresses a topic of great current interest: the role of GM in the pathogenesis of neurodegenerative disorders. The review is well structured and written. It is exhaustive despite having to analyze many topics citing the various mechanisms underlying the development of neurodegeneration. The citations are sufficient and relevant.

I would suggest, as changes to the text, an expansion of paragraphs 8 and 9 (ASD and Anxiety and Depression) which are less in-depth than the others. Furthermore, to facilitate reading, some figures could be added that emphasize the most important information both in the initial part relating to the Gut-GM-Brain Axis and in the second part of the review relating to the various neurodegenerative diseases.

Response 1 : 

We thank the Editor and the Reviewer for their constructive comments and based on their suggestions, the manuscript has been revised. The comments were very well taken and the consequent changes definitively ameliorated the quality of the manuscript. Here we are submitting a revised version of our manuscript based on their suggestions. Our detailed response to each of the Reviewers’ points is reported below.

To improve the quality of the manuscript we have, in fact, followed her/his suggestions and changes can be found in the manuscript on paragraph 8 and 9. Moreover, 2 additional figures have now been added.

8. Autism spectrum disorder (ASD)

Autism spectrum disorder (ASD) is a multifactorial neurodevelopmental disorder, influenced by genetic and environmental factors, characterized by deficits in social communication and the presence of restricted interests and repetitive behaviours, estimated to have an international prevalence of 0.76%; however, this only accounts for approximately 16% of the global child population[190].

Treatments are multiple and not uniform and further research is needed to understand behavioural and therapeutic response in ASD [191], [192]. Although the topic has recently exploded and there is not enough knowledge or a universal consensus yet, dysbiosis has been directly or indirectly associated to ASD. In particular, a reduced [193] or excessive [194], [195] abundance of lactobacilli has been reported in several studies and despite appearing contradictory[196], it suggests a clinical association between lactobacilli dysbalance and the pathogenesis of ASD. The use of probiotics in the form of supplements has also been described as an adjuvant therapeutic procedure to improve ASD-related symptoms and dysbiosis in affected children with an age spanning between 5 and 9 years old treated for 3 months with a multistrain probiotic containing Lactobacillus acidophilus, Lactobacillus rhamnosus and Bifidobacterium longum [197]. Such improvements were also observed in a child who took multi-strain supplements for 4 weeks [198]. From a neurophysiological point of view, ASD is characterized by an alteration of the GABAergic system, which is the main inhibitory neurotransmitter of the central nervous system. The atypical alternation between excitation and inhibition at the brain level affects the reactive behavior of ASD patients [199]. One study in particular has highlighted that the integration of probiotics in ASD children promotes an improvement in brain functions, attenuating the imbalance of excitation/inhibition[199]. In autistic rats, the integration of probiotics, and in particular B. longum, CCFM1077, has shown an improvement in brain activity and in the functionality of microglial cells present at the level of the cerebellum [200]. Data generated in murine models have shown that ASD symptom could improve thanks to LBP supplementation by maintaining intestinal fitness and homeostasis [201], [202]. Therefore, it seems that GM modulation could be helpful in some patients [203], as also shown in a randomized, double-blind, placebo-controlled trial in younger children within the spectrum [118]. Also, data generated in a study conducted in 35 subjects with ASD aimed to explore autonomic responses to daily LBP probiotic supplementation in combination with intra-nasal oxytocin, indicated that autonomic function indices improved [204].

9.Anxiety and depression

As in the neurodegenerative disorders discussed so far, the GM could also affect behaviour-related and psychiatric conditions [205]. As already mentioned, LBP is one of the most promising probiotic bacteria associated with CNS functions. It has been estimated that 25% of human drugs could negatively influence the GM, which could potentially even aggravate psychiatric disorders, and a large proportion of these medications are antipsychotics [206], [207]. Probiotic supplementation may be a safe and effective therapeutic strategy in psychiatric disorders including anxiety and depression [208], and the clinical use of probiotics as a therapeutic strategy to alleviate depressive and anxiety-related symptoms has been considered effective not only in patients with active psychiatric disorders, but also in healthy subjects subjected to very stressful daily life events [209]. Probiotic supplementation has been shown to increase the abundance of LBP involved in modulating psychiatric disorders and stress-induced behaviors [126], [151], [210]. Moreover, it has been shown in humans and mice that LBP strain PS128 beneficially modulates neurochemicals and stress-related symptoms, anxiety, depression-like behaviours, and major depression disorder [35], [132]. Also, a very recent study in BALB/c mice indicated that LBP APsulloc 331261 downregulated hippocampal inflammation, induced a microbial shift towards community producing the SCFAs acetate and propionate, and attenuated depressive-like behaviour [119]. Authors concluded that this modulation could stimulate an increase in colonic serotonin synthesis and neurogenesis in the brain [119]. In a double-blind, placebo-controlled trial, LBP 299v supplementation, in combination with selective serotonin reuptake inhibitors, in patients with a major depressive disorder improved cognitive performance and decrease kynurenine concentration [108].

Reviewer 2 Report

Comments and Suggestions for Authors

The manuscript entitled "Psychobiotic properties of Lactiplantibacillus plantarum in neurodegenerative diseases" provides a comprehensive review of the current knowledge on the role of gut microbiota in neurodegenerative diseases with a focus on the potential benefits of Lactiplantibacillus plantarum (LBP) strains. The manuscript largely summarizes existing studies without a critical evaluation of their methodologies, limitations, and potential biases. This lack of critical analysis makes it difficult to assess the robustness of the conclusions drawn. For instance, differences in study designs, animal models, and sample sizes are not adequately addressed, which could impact the validity of the findings. Further, this manuscript tends to overgeneralize the potential benefits of LBP across various neurodegenerative diseases without sufficient differentiation between disease-specific mechanisms and outcomes. The review predominantly focuses on the positive aspects of LBP supplementation, with minimal discussion of potential negative effects or limitations. For example, possible adverse effects, the risk of overstimulation of the immune system, or the variability in patient responses are not adequately considered. This creates a biased view that may not fully reflect the complexities of using probiotics in neurodegenerative diseases. Given the lack of critical analysis and overemphasis on positive outcomes, the manuscript’s conclusions may be misleading. The suggestion that LBP could be universally beneficial across all neurodegenerative diseases without acknowledging the complexities and nuances of each condition could lead to unrealistic expectations in the scientific and medical communities.

Comments on the Quality of English Language

Minor editing of English required

Author Response

Comments 1: 

The manuscript entitled "Psychobiotic properties of Lactiplantibacillus plantarum in neurodegenerative diseases" provides a comprehensive review of the current knowledge on the role of gut microbiota in neurodegenerative diseases with a focus on the potential benefits of Lactiplantibacillus plantarum (LBP) strains. The manuscript largely summarizes existing studies without a critical evaluation of their methodologies, limitations, and potential biases. This lack of critical analysis makes it difficult to assess the robustness of the conclusions drawn. For instance, differences in study designs, animal models, and sample sizes are not adequately addressed, which could impact the validity of the findings. Further, this manuscript tends to overgeneralize the potential benefits of LBP across various neurodegenerative diseases without sufficient differentiation between disease-specific mechanisms and outcomes. The review predominantly focuses on the positive aspects of LBP supplementation, with minimal discussion of potential negative effects or limitations. For example, possible adverse effects, the risk of overstimulation of the immune system, or the variability in patient responses are not adequately considered. This creates a biased view that may not fully reflect the complexities of using probiotics in neurodegenerative diseases. Given the lack of critical analysis and overemphasis on positive outcomes, the manuscript’s conclusions may be misleading. The suggestion that LBP could be universally beneficial across all neurodegenerative diseases without acknowledging the complexities and nuances of each condition could lead to unrealistic expectations in the scientific and medical communities.

Response 1 : 

We thank the Reviewer for her/his insightful comment and agree with her/his point of view. We have now substantially extended the conclusion paragraph addressing and summarizing better the potential pitfalls and limitations of the reported studies, making our critical view clearer.

10.Conclusions, limitations and future perspective

In recent years, many evidence emerged from humans and rodent models pointing at the role of the GM in the physiology of the CNS. Dysbiosis, in fact, has been strongly suggested to play a crucial part in the onset and development of neurodegeneration. In this review, we have discussed the role of GM in health and disease of the CNS, and in particular the psychobiotic potential of LBP in neurodegenerative disorders. Thanks to its ability of beneficially modulating the GM composition, by promoting the enrichment of SCFAs-producers, and its capacity of boosting the production of neurotransmitters, LBP supports the homeostasis of the gut-brain axis and regulates anti-inflammatory and anti-oxidant pathways in the CNS. All the emerging data are also laying the foundation for the design of novel therapeutic or adjuvant strategies for the clinical management of patients affected by neurodegenerative diseases. However, it is worth pointing out that we are far to make definitive conclusions and caution should be taken in prescribing probiotic therapy in an indiscriminate way, as some groups have reported negative effects spanning from immunoreactivity to sepsis and even antibiotic resistance [211].This should be considered especially in vulnerable subgroups, such as elderlies and immunocompromised and/or critically ill patients[212]. Moreover, we should be careful in translating data generated in animal models to humans and, even though data generated by RCT are emerging, there are important pitfalls to take into account, such as length of treatment, strain of probiotics used, and dosage; moreover, patients’ recruitment in different RCT, even within the same neurodegenerative disorder, are not optimal yet. In addition, to have more reliable data on intestinal colonization and benefits in patients’ physiology regarding modulation of their GM, formulation of probiotics administered to patients should be standardized to rely on the bioavailability of each strain or multistrains. Also, the definition of “healthy microbiome” is still one of the most enigmatic issues, more data on the role of diet-dependent GM changes in health and disease are needed, the interaction between drugs and the GM needs further studies, and there is no one-size-fit-all solution to target neurodegenerative disorders, despite the promising results. Diet and lifestyle of patients involved in these types of studies should always be considered, as it may influence intestinal colonization. For example, in AD, probiotics do not seems to establish colonization in a stable gut milieu, therefore it becomes almost impossible to understand which particular bacteria in a multistrain formulation can be the most impactful [213]; moreover, in some AD patients serotonin syndrome in subjects can occur [214]. Therefore, clinical trials design must be implemented, extending the number of recruited patients and foreseeing longer follow-up; furthermore, more endpoints should be included such as inflammatory biomarkers and specific cognitive assessment related to each neurodegenerative disease. Anyhow, the integration of multiomic data and an approach based on system biology will definitely move the field forwards and more advanced data interpretation will allow a deeper comprehension of the conundrum gut-microbiota-brain axis.

Round 2

Reviewer 2 Report

Comments and Suggestions for Authors

These revisions have strengthened the manuscript and ensured that it meets the standards required for publication.